# The Use of a Cooperative-Learning Activity with University Students: A Gender Experience

**Salvador Baena-Morales [1]**, **Daniel Jerez-Mayorga [2]**, **Francisco Tomás Fernández-González [3]** and **Juan López-Morales [4],***

1 Department of General Didactics and Specific Didactics, Faculty of Education, University of Alicante, 03690 Alicante, Spain; salvador.baena@ua.es
2 Faculty of Rehabilitation Sciences, Universidad Andres Bello, 7591538 Santiago, Chile; daniel.jerez@unab.cl
3 Department of Physical Activity and Sport Sciences, Pontifical University of Comillas (Centro de Estudios Superiores Alberta Giménez), 07013 Palma, Spain; francis.gonzalez.fernandez@gmail.com
4 Department of Social Anthropology, Faculty of Philosophy and Letters, University of Granada, 18071 Granada, Spain
* Correspondence: jlopezmorales@ugr.es

**Abstract:** The UN Sustainable Development Goals (SDG) show how education is essential for creating values in students. In particular, SDG 4 (quality education) and SDG 5 (gender equality) indicate how co-education should be a sustainable benchmark. Co-educational methodologies have been studied for decades. Among them, cooperative learning is considered a valid technique for developing social relations and competences. This study aims to describe and characterize the gender differences between university students regarding their impressions and behaviors when working cooperatively. One hundred and seventy-seven university students (98 women and 79 men), from Physical Education and Primary Education degree courses, worked with Aronson's Jigsaw technique. After its completion, they completed a questionnaire to analyze cooperative work in higher education (ACOES). The results are organized into seven dimensions. The main gender differences found show that women gave a higher evaluation to relating cooperative learning to future teaching roles ($p = 0.017$) and to understanding the need for cooperative tasks ($p = 0.035$). Additionally, female students prefer groups to be organized according to academic criteria and that they should remain stable throughout the academic period. Both genders value Aronson's Jigsaw as a good method for developing social competences, although they are more neutral when considering it effective at improving academic performance. These findings help to generate a gender-cooperation profile that will enable future research to discuss results more accurately.

**Keywords:** methodology; co-education; higher education; cooperative learning

## 1. Introduction

The United Nations (UN) General Assembly adopted the 2030 Agenda for sustainable development. It consists of an action plan in favor of people, the planet, and prosperity, and promotes universal peace and access to justice [1]. It sets forth 17 objectives, four of which are relevant to this article. Sustainable Development Goal (SDG) 4 seeks to "guarantee inclusive, equitable and quality education and promote lifelong learning opportunities for all". It is intended to ensure free education and equal access for all children. Goal 5 seeks to achieve gender equality and empower all women and girls. Despite advances in equality, there are still many women with difficulties due to discriminatory laws and social norms. SDG 10 seeks to reduce inequality within and between countries, while SDG 11 seeks to make cities more inclusive, safe, resilient, and sustainable [1–3].

The new educational demands are based on teaching and learning processes that favor connecting context and content, considering them as an interdependent whole [4]. This paradigm entails the appearance of new teaching and learning methods that involve an increase in collaboration for the acquisition of new knowledge and skills [5]. In this sense, cooperative learning (CL) is a pedagogical practice that has been used for decades at different educational levels [6], and, in the previous literature, has been suggested as an effective method to improve learning performance in students [7–9]. Therefore, knowledge of the variables that could influence behaviors and results in a cooperative group should be studied. For example, CL is characterized by the ability to develop sociability skills synergistically in learning [10,11]. This is due to the development of the group members' interaction, since they must physically work together in the completion of a task [12]. Furthermore, CL promotes positive interdependence and involves a process where decisions are shared, discussed, and agreed upon [13]. These characteristics consequently result in a positive relationship of interdependence between the students, the content, and the context [14].

Despite these characteristics, in the university context, it has been found that most group practice proposals are based on collaboration and not on cooperation [15]. However, collaborative group work does not seem to offer guarantees of effectiveness in improving interactions and learning; neither is equal participation within the workgroup guaranteed [16]. To improve this type of practice, CL is supported by a theoretical foundation that has been validated by research and corroborated by teachers [6]. It has also been shown to develop co-responsibility in learning [13]. Therefore, the significant difference with traditional group work based on collaboration is that each member of the group has to develop responsibility which helps to achieve group success [17]. Díaz-Aguado [18] emphasizes that these intrinsic characteristics of CL could improve students' motivation through a context that enhances relations between peers and teachers. Regarding this idea, during the lockdown period caused by the coronavirus, teachers even attempted to carry out cooperative techniques in a virtual way to promote social habits during the lockdown [19]. Therefore, CL can enable students to participate in the teaching–learning process, which not only allows students to learn specific content, but also helps their classmates to acquire it, too [6,12,16,17].

## 1.1. Aronson's Jigsaw as a Cooperative Technique

Various different methods and techniques are used in CL. One of the ten most used techniques used today, according to Johnson and Johnson [6], is Aronson's Jigsaw (AJ) [20]. Velázquez [13] reinforces this statement by asserting that AJ is one of the most used among teachers, while Fernández-Rio [21] considers it the best known and most used CL technique, although some authors consider it a CL technique very similar to others, such as Problem-Based Learning. However, Leyva-Moral and Riu [5] indicate that AJ has its own identity as a technique. AJ is based on the division of the content to be learned into different parts. In this way, each member of the group will study and specialize in one part, to subsequently explain it to other classmates who will also have specialized in another "piece of the puzzle" [22]. Each group member is thereby responsible for learning part of the content, becoming an expert on that specific sub-content. Thus, in this information exchange process, each student develops a role that makes them an indispensable part of the learning process; and with their contribution, they make it possible to achieve the assigned task. Therefore, AJ promotes positive interdependence between the group members and responsibility [17,23].

Leyva Moral and Riu [5] state that the AJ cooperative technique has been used successfully within the university environment in disciplines such as computer science [24], medicine [25,26], dentistry [23], and sports science [27]. These investigations mainly confirm the improvement of students' perceptions; however, they do not confirm that learning is greater with AJ. In this regard, they point out that AJ was not a highly valued technique among nursing students, who questioned whether it produced greater learning compared to traditional methods [5]. In addition, Bratt [28], despite highlighting the essential characteristics of CL, such as improvement in group identity or attitudes towards learning, points out that it does not seem to produce a feeling of progress in learning acquisition. Schoenecker [29] points

out the importance of knowing AJ's formative virtues and not conceiving it as a simple sum of parts. In previous research, the students' feelings concerning CL's different characteristics have been evaluated, identifying the coordination of tasks and the irresponsibility of other group members as the main problems [30]. Fernández-Rio [21] asserts that both the teaching staff and the students tend not to be prepared to implement AJ correctly. The author also notes that the technique often fails when integrating the learning content transmitted by the person in charge. Additionally, a certain reluctance to change the teaching role and the teacher's loss of control have been detected [21] because, through this technique, the group members depend on each other to learn the content to be worked on. In this way, it is unquestionable that the performance of each member of the group is essential to achieving the common goal, thus interpersonal relationships are fostered and personalities adapt to the requirements of the context during the cooperation process. However, little research has yet been undertaken to examine whether there is a difference between the perceptions of male and female students in this cooperative context created by AJ.

### 1.2. Gender and Cooperative Learning

SDG 5, on gender equality, explains in its article 5.5 that women should be ensured effective participation and equal opportunities for leadership at all levels of decision-making [1,2]. Additionally, SDG 4 stipulates in its article 4.7 that all learners should acquire the knowledge and skills needed to promote sustainable development, including gender equality [2]. Hence, gender stereotypes are among the main challenges for gender equality in the sustainable development goals [31]. In addition, inclusive models in education must be based on parameters of equality, with CL being a scenario that could promote opportunities in an equitable way and encourage equality before the achievement of success [8,16,32]. This egalitarian context will help bring about a decisive improvement in student learning [21]. To successfully apply CL, Putnam [33] states that the groups must be heterogeneous, meaning that teachers need to adapt and personalize both the criteria for success and their expectations or requirements of the tasks to the needs and abilities of every student in the team. In addition, Kagan [16] shows how techniques that structure the different activities should be applied to guarantee the equal or equitable participation of the entire group, since this does not arise spontaneously from the students. If these basic rules are taken care of, CL will enable a high degree of equality, in the sense that there must be a degree of symmetry in the roles undertaken by participants in a group activity [34]. For all these reasons, teachers must know the necessary information to correctly apply CL techniques [6].

In terms of gender, the scientific evidence shows that men and women have different senses of self. Men have a more autonomous and separate perception of the self, while women seem to perceive themselves in a relational way based on connections and relationships [35]. In addition, traditional studies indicated a trend where women had more difficulties in performing tasks in environments perceived as more competitive. Conversely, if men perceived a competitive scenario, they increased their capacity. This circumstance could be explained by the fact that in a competitive environment, women tended to focus more on interpersonal aspects, while men focused on aspects related to the competitive achievement [36]. However, male pairings have been shown to have the lowest expectations and cooperation rates [37]. In addition, women have greater affiliation, cooperative attitude, and interdependence [35]. All this evidence shows different behaviors associated with gender in environments that require cooperation. Despite these characteristics related to gender and CL, gender inequality has consequences that are transferable to social interactions [38,39]. We need to examine and improve our knowledge about the variables implicit in gender that can condition the perceptions of CL.

## 2. Objective and Methods

### 2.1. Objective

This study aimed to examine and show the differences in gender perceptions of university students after cooperative work. We hypothesized that CL would be a positively valued technique for promoting social competencies for women. Additionally, we expected female students to have more structured behavior in carrying out cooperative tasks.

### 2.2. Population and Sample

The present research was a cross-sectional study conducted through a self-administered questionnaire. A total of 177 university students, Physical Education and Primary Degree of University of Alicante, 98 female students (age = 22.29 ± 5.61 years) and 79 male students (age = 22.91 ± 5.17 years). The inclusion criterion to participate in the study was to have previously carried out an exercise where the use of CL as a cooperative technique was presented and practiced.

### 2.3. Questionnaire

The instrument used was the Questionnaire for the Analysis of Cooperation in Higher Education (ACOES Scale) due to its acceptable internal consistency and its capability for item discrimination. This questionnaire is based on a Likert Scale (1 = Totally disagree, and 5 = Totally agree). This scale aims to analyze group work in higher education students. It enables us to examine different aspects of CL (student understanding, usefulness, and intergroup norms, among others). It is made up of 49 items and grouped into seven dimensions (Table 1) [40]. It was therefore adopted to evaluate cooperative work through different dimensions such as the understanding of group work, the usefulness of cooperative work for their training, planning of the work of the groups by teachers, criteria for organizing groups, group rules, internal functioning of the groups, and the effectiveness of group work. The dimensions of the questionnaire have different objectives (Figure 1):

A. Understanding of group work. The aim of this dimension is to understand the mental representation and the meaning of the CL for students.
B. Usefulness of cooperative work for their training. It analyses the validity of the CL for the students and whether it is perceived as useful for improving social relations, learning, or future professional work.
C. Planning of the work of the groups by the teachers. To learn the opinion of the students regarding the quantity, complexity, coordination, and teaching of the cooperative work.
D. Criteria for organizing the groups. This examines which criteria are taken into account to form the work teams. It is asked whether they should be organized for personal or academic reasons, temporary stability, or their homogeneity. In addition, the number of members and the respect for different roles are evaluated in this dimension.
E. Group rules. The internal regulation of the group is the factor evaluated in this dimension. The aim is to find out the importance of establishing operating rules or whether these should be internal to the group (students), external (teachers), or mixed.
F. Internal functioning of the groups. The objective is to learn the tasks that are carried out by the students before the end of the presentation related to CL.
G. Effectiveness of group work. This dimension aims to evaluate the cooperative conditions in which better levels of performance and production are produced. For this purpose, the items ask about the weighting of group work in the final qualification, and the inclusion of self-evaluation or co-evaluation, among others.

**Table 1.** Results obtained by gender in the Analysis of Cooperation in Higher Education (ACOES) questionnaire.

| Items | Women (n = 108) | Men (n = 69) | p-value | ES | Interpretation |
|---|---|---|---|---|---|
| **I CONSIDER THAT GROUP WORK IS: (CON)** | Median (5–95 percentile) | Median (5–95 percentile) | p-value | ES | Interpretation |
| A good method to develop my social skills: Argumentation, dialogue, listening skills, debate, respect for dissenting opinions | 3 (5–5) | 3 (5–5) | 0.365 | 0.071 | Irrelevant |
| An opportunity to get to know my peers better | 3 (5–5) | 3 (5–5) | 0.874 | 0.012 | Irrelevant |
| A way to better understand knowledge | 2 (4–5) | 2 (4–5) | 0.992 | 0.000 | Irrelevant |
| A way to share the total workload | 2 (4–5) | 3 (4–5) | 0.378 | −0.073 | Irrelevant |
| A way to make test preparation easier | 2 (4–5) | 2 (4–5) | 0.729 | 0.030 | Irrelevant |
| Dimension Mean | | | 0.903 | | |
| **PERSONALLY, GROUP WORK HELPS ME TO: (1AYU)** | Median (5–95 percentile) | Median (5–95 percentile) | p-value | ES | Interpretation |
| Expose and defend my ideas and knowledge to other people | 4 (3–5) | 3 (4–5) | 0.880 | 0.011 | Irrelevant |
| Feel an active part of my own learning process | 3 (4.5–5) | 3 (4–5) | 0.080 | 0.140 | Small |
| Understand the knowledge and ideas of peers | 3 (5–5) | 3 (4–5) | 0.124 | 0.124 | Small |
| Understand the importance of coordinated work in my professional future as a teacher | 3 (5–5) | 3 (4–5) | 0.017 * | 0.187 | Small |
| Reach agreements in the face of different opinions | 3 (4.50–5) | 3 (4–5) | 0.646 | 0.036 | Irrelevant |
| Search for information, research and learn autonomously | 3 (4–5) | 2 (4–5) | 0.058 | 0.158 | Small |
| Dimension Mean | | | 0.043 * | | |
| **ABOUT THE PLANNING THAT THE TEACHERS DO FOR THE GROUP WORK, I THINK THAT: (PLA)** | Median (5–95 percentile) | Median (5–95 percentile) | p-value | ES | Interpretation |
| The amount of group work requested is adapted to the course load | 1 (3–5) | 1.40 (4–5) | 0.103 | −0.139 | Small |
| The level of difficulty of the group work is appropriate for our training | 2 (4–5) | 3 (4–5) | 0.054 | −0.160 | Small |
| There is coordination between the group work requested in the different subjects | 1 (3–4.65) | 1 (2–5) | 0.217 | −0.108 | Small |
| Attending practical classes solves the doubts that I have when preparing the group work | 2 (4–5) | 2.40 (4–5) | 0.541 | −0.050 | Irrelevant |
| Dimension Mean | | | 0.327 | | |
| **THE CONSTITUTION OF THE GROUP MUST: (ORG)** | Median (5–95 percentile) | Median (5–95 percentile) | p-value | ES | Interpretation |
| Be carried out by the students applying friendship criteria | 1 (4–5) | 1 (3–5) | 0.992 | 0.000 | Irrelevant |
| Be carried out by the students applying academic criteria | 1 (4–5) | 1 (3–5) | 0.004 * | 0.244 | Small |
| Be carried out by the teaching staff applying academic criteria | 1 (2.5–5) | 1 (3–5) | 0.624 | −0.422 | Medium |
| Have a diverse composition of group members (age, sex, education, experience) | 2 (4–5) | 2 (4–5) | 0.453 | 0.063 | Irrelevant |
| Be stable throughout the course, term, and year | 2.35 (4–5) | 1.4 (4–5) | 0.023 * | 0.190 | Small |
| Be modified to perform different activities in the same subject | 1 (3–5) | 1 (3–5) | 0.033 * | −0.184 | Small |
| Incorporate the appointment of a group coordinator | 3 (3–5) | 3 (4–5) | 0.413 | 0.069 | Irrelevant |
| Dimension Mean | | | 0.486 | | |

**Table 1.** *Cont.*

| Items | Women (*n* = 108) | Men (*n* = 69) | | | |
|---|---|---|---|---|---|
| **THE GROUP'S OPERATING RULES: (FUN)** | Median (5–95 percentile) | Median (5–95 percentile) | *p*-value | ES | Interpretation |
| There should be no rules | 1 (1–3) | 1 (2–4) | 0.155 | −0.115 | Small |
| There must be rules, but established by the students | 2.35 (4–5) | 1.40 (4–5) | 0.065 | 0.159 | Small |
| There must be rules, but established by the faculty | 1 (3–4) | 1 (3–5) | 0.128 | −0.128 | Small |
| They must be negotiated between teachers and students | 1 (5–5) | 2 (5–5) | 0.590 | −0.043 | Irrelevant |
| They must be included in a document where the responsibilities assumed by the group are specified | 1 (4–5) | 1 (4–5) | 0.871 | 0.013 | Irrelevant |
| They must define the roles that each of the people who make up the group will play | 1 (4–5) | 1.4 (4–5) | 0.845 | 0.016 | Irrelevant |
| They should include consequences for participants for not fulfilling the commitments made | 3 (4–5) | 2 (4–5) | 0.728 | 0.029 | Irrelevant |
| They must specify the time and place of the meetings | 1 (4–5) | 1 (3–5) | 0.136 | 0.129 | Small |
| Must include mandatory attendance at meetings | 2 (4–5) | 2 (4–5) | 0.9854 | 0.015 | Irrelevant |
| Dimension Mean | | | 0.970 | | |
| **USUALLY, WHEN DOING GROUP WORK: (2FUN-I)** | Median (5–95 percentile) | Median (5–95 percentile) | *p*-value | ES | Interpretation |
| We meet at the beginning to plan the different steps we have to take | 2 (5–5) | 3 (4–5) | 0.921 | 0.007 | Irrelevant |
| We consult the basic documentation provided by the teacher | 3 (5–5) | 3 (4–5) | 0.001 * | 0.257 | Small |
| We search for information in different sources (internet, library) | 3 (5–5) | 2.4 (4–5) | 0.012 * | 0.197 | Small |
| We make decisions, in a consensual way, to guarantee the overall coherence of the group work | 3 (5–5) | 3 (4–5) | 0.031 * | 0.172 | Small |
| When carrying out the work, we have discussions so that the whole group knows what the others are doing and we have a good idea of the progress of the activity | 2.35 (5–5) | 3 (4–5) | 0.543 | 0.049 | Irrelevant |
| All members of the group participate equally | 1 (4–5) | 1.4 (4–5) | 0.225 | 0.105 | Small |
| We evaluate it and make proposals for improvement | 2 (4–5) | 1.4 (4–5) | 0.176 | 0.114 | Small |
| Dimension Mean | | | 0.035 * | | |
| **THE GROUP'S PERFORMANCE IMPROVES IF: (REN)** | Median (5–95 percentile) | Median (5–95 percentile) | *p*-value | ES | Interpretation |
| The teachers provide clear guidelines for the group activities to be developed | 3 (5–5) | 2 (4–5) | 0.037 * | 0.163 | Small |
| The activities proposed by the teachers require analysis, debate, reflection, and criticism | 2 (4–5) | 2 (4–5) | 0.257 | −0.097 | Irrelevant |
| The teaching staff supervises the work of the group | 2 (4–5) | 1.4 (5–5) | 0.083 | −0.146 | Small |
| Teachers control regular class attendance | 1 (4–5) | 2 (4–5) | 0.293 | 0.089 | Irrelevant |
| The work is adequately valued in the overall grading of the subject | 3 (4–5) | 2.4 (4–5) | 0.464 | −0.060 | Irrelevant |
| The teaching staff informs us in advance about the evaluation criteria of the group activity | 2 (5–5) | 2.5 (5–5) | 0.785 | 0.022 | Irrelevant |
| Teachers evaluate the different levels of participation of each of the group members | 1 (4–5) | 1 (3–5) | 0.555 | 0.051 | Irrelevant |
| Each student's self-evaluation is incorporated into the group's overall assessment | 1 (4–5) | 1 (4–5) | 0.655 | −0.038 | Irrelevant |
| The members of the group evaluate each other | 1 (3–5) | 1 (4–5) | 0.087 | −0.152 | Small |
| The teaching staff gives the group work significant weight in the final grade of the course | 2 (4–5) | 1.4 (4–5) | 0.457 | −0.062 | Irrelevant |
| Dimension Mean | | | 0.619 | | |

Data are presented as median (5–95 percentile), Effect Size (ES) is presented as Rank–Biserial Correlation, * *p* < 0.05.

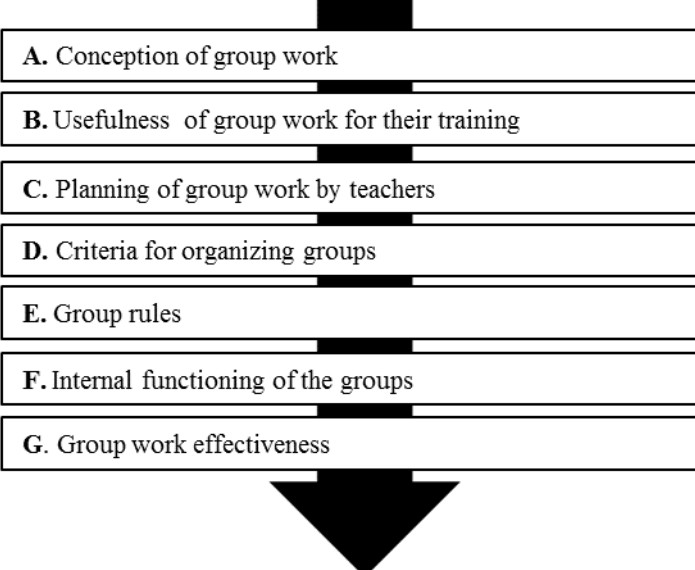

**Figure 1.** Dimensions studied in the questionnaire.

## 2.4. Data Collection and Analysis Procedure

Four students who had not previously carried out the exercise were excluded from the study. Subsequently, the participants were evaluated on the dynamics of the task to confirm that they understood the process. A series of protocols were followed to obtain the maximum response rate and the greatest sincerity in the completion of the questionnaire. During completion of the exercise, the researchers did not intervene, but only explained the process. The groups were assigned randomly, trying to avoid grouping by affinity or interest. The number of members per group was 4 or 5, respecting that all the students had a similar amount of content. The researchers stressed the anonymity of the questionnaire results. The questionnaire was completed directly after the end of the AJ task, preventing any comments or discussion among the students in order not to condition the responses. A graphic representation of the procedure is summarized in Figure 2.

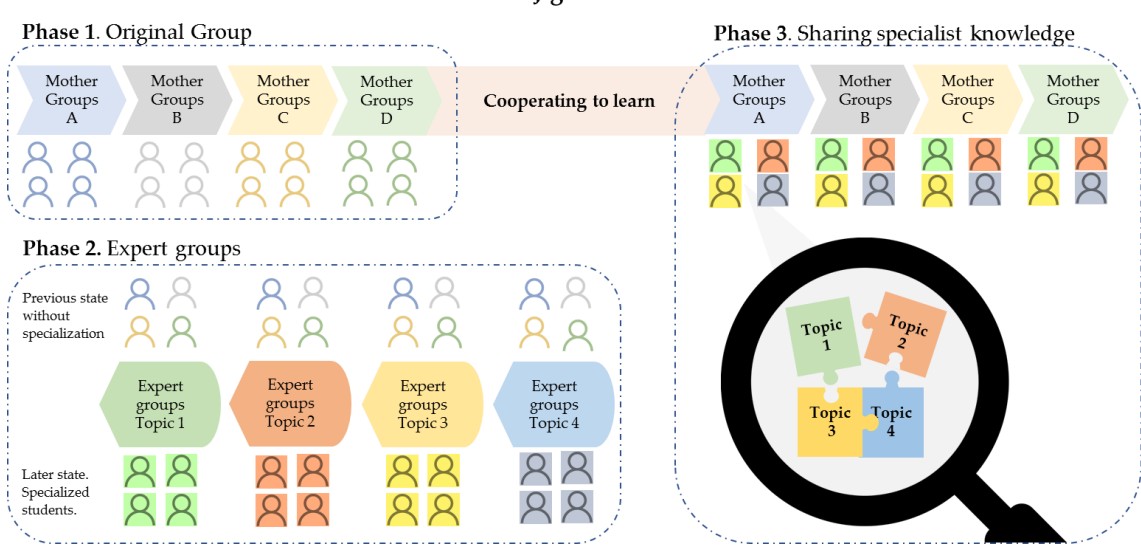

**Figure 2.** Cooperative work protocol implemented through the Jigsaw Method.

The data analysis is presented as median (mdn) and 5–95 percentiles. The difference between evaluations of the collaborative work and gender was analyzed through the Mann–Whitney U test. The data processing was carried out with JASP Software (Version 11.1.0).

## 3. Results

The results associated with the dimension "personally, group work helps me to" present significant differences between women (mdn = 5) and men (mdn = 4), $p = 0.017$, in the item "Understanding the importance of coordinated work in my professional future as a teacher".

In the dimension, "the constitution of the group should", the items "be carried out by the students applying academic criteria" and "be stable throughout the course, term, year" present significant differences between women (mdn = 4) versus men (mdn = 3), $p = 0.004$ and 0.023, respectively. In turn, in the item "be modified to perform different activities in the same subject", there are differences between men and women ($p = 0.033$).

Similarly, in the dimension "Usually, when doing group work", the items "We consult the basic documentation provided by the teacher", "We search for information from different sources", and "We make decisions, by consensus, to ensure the overall consistency of the group work" present differences between women (mdn = 5) compared to men (mdn = 4).

Finally, in the dimension "group performance improves if", the women are in total agreement (mdn = 5) with the item if "teachers provide clear guidelines for group activities to be developed", $p = 0.037$, in contrast to the men (mdn = 4) (Table 1).

The data analysis is presented as median (mdn) and 5–95 percentiles. The difference between evaluations of the collaborative work and gender was analyzed through the Mann–Whitney U test. The effect size (ES) was determined through Rank–Biserial Correlation [41]. The criteria to interpret the magnitude of the ES were as follows: Irrelevant (<0.1), small (0.1–0.30), medium (0.30–0.50), and large (>0.50) [42]. The data processing was carried out with JASP Software (Version 11.1.0).

## 4. Discussion

The aim of this study was to identify the gender differences in the perceptions of university students following the use of a cooperative technique. The results obtained make it possible to establish a gender profile with regard to the use of CL. Female university students (FU) register significant differences in a greater number of variables than men. Regarding the usefulness of cooperative work, FU relate it more to their future professional teaching career than men. Additionally, with regard to the organization of cooperative groups, the FU consider that their design should be done by the students, respecting academic criteria and remaining stable throughout the course. FU also place greater value on the procedure that takes place during CL, making greater use of documentation provided by teachers, using different sources of information, and respecting consensus in decision-making to provide overall consistency. Finally, FU highlight the importance of clarity in the rules imposed by teachers in order to achieve greater performance in the cooperative task.

As far as we know, this is the first study to analyze the differences between genders in perceptions of CL using AJ. The data allow us to increase our knowledge in these areas to find out possible differences in the behavior of university women and men when they do cooperative work.

One of the main findings of this study shows how, in relation to the additional help provided by cooperative techniques, there are significant differences for women ($p = 0.04$). This dimension allows us to determine the students' assessment of the usefulness of group work to enhance their social interactions, autonomous learning, and future professional performance. Thus, we observed that the FU were more confident in valuing positively the transfer of these cooperative skills to the workplace. Although Matzumura-Kasano et al. [43] did not find differences in this dimension for gender, this aspect is in agreement with Lerís et al. [44], who indicate that CL, beyond facilitating academic learning, prepares the student for future work performance. As discussed, male university students (MU) give less value to the transfer of CL to working life, which according to Cigarini [37],

could be explained by the fact that when men are paired in a cooperative task, they show a lower rate of cooperation and expectations about the success of the task. Therefore, the results obtained demonstrate that women value CL significantly more as a technique to help them understand the importance of coordinated work for their professional future.

Another noteworthy result is related to the FUN-I dimension. This dimension aimed to clarify the actions taken in the group work process prior to its completion. The results showed that FU value all the tasks prior to the end of the activity ($p = 0.035$) significantly more than the MU. Specifically, the FU are more careful in the process of consulting the documentation provided by the teachers (p = 0.001), searching different sources of information ($p = 0.012$), and taking decisions in a consensual way, in order to guarantee the final coherence of the cooperative work ($p = 0.031$). In this regard, it should be noted that the type of task carried out has been documented as an essential factor for determining gender involvement in CL [45].

The purpose of one of the dimensions we analyzed was to generate a profile of the university student when comprehending cooperative work. The sample analyzed shows no gender differences for this dimension. Therefore, we can determine that male and female university students have the same conception of the usefulness of cooperative work. In this regard, both genders coincide in valuing the cooperative capacities of AJ, seeing it as a good method for developing social skills [46], getting to know colleagues better [47], and sharing the workload [17]. However, the items related to the acquisition of knowledge score below 4 in both cases, with both FU and MU doubting that the cooperative technique is an efficient method for preparing exams (girls = 3.5/5 and boys 3.6/5) or that it is an opportunity to better understand the facts (girls = 3.9/5 and boys 3.9/5). These results are very similar to those obtained by Matzumura-Kasano et al. [43], in which medical students did not value CL as an optimal technique for exam preparation (3.7 ± 1.1). Despite being positive scores, close to four points out of five, it worth noting how there is a decrease in the score in the items that evaluated the learning potential. These results could be explained by the lack of relationship between the way the contents were evaluated and the way they were prepared. However, the data obtained seem to be in conflict with the conclusions obtained by Navarro [27], who stated that CL is a documentation and research process that results in better academic performance in more than 90% of the students examined. This perception could be an important factor, as an improvement in their learning has been described when students perceive CL positively [9].

The organization and constitution of the groups is another of the problems detected with respect to CL. The variety of variables that determine the constitution of a group makes knowledge of this dimension particularly important. Personal criteria, group stability and the homogeneity/heterogeneity of the members have been taken into account in the design of the ACOES questionnaire [40]. The results obtained show how the FU prefer that groups be created exclusively by academic criteria ($p = 0.004$). Furthermore, FU prefer the groups to be stable throughout the academic period ($p = 0.023$). In contrast, the MU consider that the cooperative groups should vary depending on the activities carried out ($p = 0.033$). Previous studies do not highlight gender differences in the creation of cooperative groups, but they do emphasize the importance of taking into account which people had previously worked together [48]. In terms of respect, a recent study shows positive student perceptions regarding interpersonal relationships [9]. With the results obtained, we can state that the FU differ from the MU in preferring more stable cooperative groups and in thinking they should be organized according to academic and not only personal criteria.

Finally, there are gender differences in relation to the variables that may condition a better performance of the cooperative group. Specifically, FU differ significantly with MU in valuing the importance of establishing previous guidelines for carrying out the activities ($p = 0.037$). These results do not coincide with those presented by Matzumura-Kasano [43], which, based on Mora-Vicaroli and Hooper-Simpson [49], emphasize the responsibility of teachers to supervise internal behaviors and confirm that the operating rules are respected. On the other hand, it should be noted that no gender differences have been found in the population studied regarding the dimensions of the group's

operating rules. This suggests that both genders have the same idea of which behavioral rules should govern cooperation. The students and teachers agree on establishing rules together and on showing how these should be reflected in documents or meetings. Another dimension that does not highlight significant differences between genders is that of planning, which must be carried out by teachers. It should be noted that this dimension depends not so much on the characteristics of CL, but on the personality of the teachers. Thus, if the questionnaire is carried out at a time when the students have many tasks to do or a heavy workload from other subjects, the results obtained could be affected. This last example could indicate a potential limitation of the present study, since the emotional state of the subjects was not jointly assessed to ensure that the responses to the questionnaire were not influenced by work overload or other personal factors. In addition, it is noteworthy that the sample analyzed came from different university degrees, which may allow the results to be extrapolated to the general university population.

## 5. Conclusions

The data analysis has allowed us to observe how women value the relationship between cooperative techniques and future professional teaching practice more positively. Furthermore, the female students considered all the tasks prior to the completion of the cooperative project—for example, consulting the documentation provided by the professors, performing searches, or making decisions in a consensual manner to guarantee the coherence of the final work—to be essential.

Another of the main conclusions obtained concerns the organization of the cooperative groups. While the female students preferred the groups to be organized exclusively by academic criteria and to be stable throughout the academic period, the university men considered that the groups should be changed according to the activities or assignments proposed.

However, we can state that both women and men have a similar perception of the usefulness of cooperative work, highlighting it as an effective technique for developing social skills, but not so much as an opportunity to improve academic performance.

The data presented will serve as a reference for future research to analyze the variables that can influence the implementation of cooperative strategies. In addition, we can state that in general terms, university students perceive Aronson's Jigsaw technique as an aid to promoting social and personal skills, but do not see it is an effective method for improving learning.

**Author Contributions:** Conceptualization, S.B.-M. and J.L.-M. methodology, S.B.-M. and F.T.F.-G.; software, D.J.-M.; formal analysis, D.J.-M.; investigation, S.B.-M.; writing—original draft preparation, S.B.-M. and J.L.-M.; writing—review and editing, S.B.-M., J.L.-M.; supervision, S.B.-M. and J.L.-M. All authors have read and agreed to the published version of the manuscript.

**Funding:** This research received no external funding.

**Conflicts of Interest:** The authors declare no conflict of interest.

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
