# Peer review of "The Use of a Cooperative-Learning Activity with University Students: A Gender Experience"

_sustainability, doi:10.3390/su12219292_

Round 1

Reviewer 1 Report

I have tried to read this ms. I have stopped at page 3. The English is a disaster! A nightmare.

Author Response

The article has been reviewed by a professional English proofreader (we attach Certificate of profesional proofreading).

Reviewer 2 Report

Line 24 - why is there letter "y" after the bracket?

Line 171 - what does it mean PA Y in the sentence with no explanation of the abbreviation before? Probably PA means Puzzle Aronson but check all abbreviation in the whole text!

Explanation of Figure 1 - where activity C disappeared? it is on the left picture but not in the right picture. Can you briefly explain or describe the figure?

Line 193 - I cannot find Table 2 in the text.

Can you think about better presentation of your questionnaire?

Line 218 - what your abbreviation CA means? No explanation is there.

Author Response

We appreciate your proposals. We attach a document responding to your proposals.

Round 2

Reviewer 1 Report

Some comments

L57: What’s the exact difference between “collaboration” and “cooperation”?

L139: The heading should be: “Objective and Method”.

L146-150: More information regarding the procedure should be provided. How were the students contacted? By whom? How many were contacted and how many agreed to participate? What were the reasons for not participating? Was it a paper and pen questionnaire? How much time did it take to answer the questions?

L153: Provide information on “internal consistency” and “capability for item discrimination”

L154: What is “an ad hoc Likert Scale”?

L157: What does “is formed by one grouped” mean?

L158: What do you mean by “therefore”?

L156-157 are almost the same as L157-158.

Figure 2: Reverse the order G and F.

L186: How many were excluded?

L188: What “protocols”?

Figure 1: I do not understand this figure.

L198: The data analysis is presented as median and percentiles: I don’t understand this.

Table 1: A huge table is presented, with P5, P50 and P95, while nothing is said about these values. Why present them? On the other hand, a comparison of median values is made, but the median values are NOT in the table. To me a p-value does not mean very much. You should also present an Effect Size, which gives a indication of the strength of the association/difference. In addition, why did you not perform some kind of factor analyses, which probably presents a much compacter indication of the differences.

L232: “The data allow us to create new research areas”: what do you mean?

L235: Where does p = 0.04 come from?

L249: Where does p = 0.035 come from?

L303: The fact that the sample came from different university degrees (?), does certainly not allow us the extrapolate the results!  

Author Response

We are grateful for the comments/suggestions. This review has helped to strengthen our work. We attach a document with the changes.
